# COVID-19 healthcare demand projections: Arizona

**Esma S. Gel**[1], **Megan Jehn**[2], **Timothy Lant**[3]*, **Anna R. Muldoon**[4], **Trisalyn Nelson**[5], **Heather M. Ross**[4,6]

1 School of Computing, Informatics and Decision Systems Engineering, Arizona State University, Tempe, AZ, United States of America, 2 School of Human Evolution and Social Change, Arizona State University, Tempe, AZ, United States of America, 3 Arizona State University, Tempe, AZ, United States of America, 4 School for the Future of Innovation in Society, Arizona State University, Tempe, AZ, United States of America, 5 School of Geographical Sciences and Urban Planning, Arizona State University, Tempe, AZ, United States of America, 6 Edson College of Nursing and Health Innovation, Arizona State University, Phoenix, AZ, United States of America

⊜ These authors contributed equally to this work.

* tim.lant@asu.edu

**Data Availability Statement:** All case data, hospitalization, data, and death data are held in a public repository at the Arizona Department of Health Resources site at https://www.azdhs.gov/preparedness/epidemiology-disease-control/

## Abstract

Beginning in March 2020, the United States emerged as the global epicenter for COVID-19 cases with little to guide policy response in the absence of extensive data available for reliable epidemiological modeling in the early phases of the pandemic. In the ensuing weeks, American jurisdictions attempted to manage disease spread on a regional basis using non-pharmaceutical interventions (i.e., social distancing), as uneven disease burden across the expansive geography of the United States exerted different implications for policy management in different regions. While Arizona policymakers relied initially on state-by-state national modeling projections from different groups outside of the state, we sought to create a state-specific model using a mathematical framework that ties disease surveillance with the future burden on Arizona's healthcare system. Our framework uses a compartmental system dynamics model using a SEIRD framework that accounts for multiple types of disease manifestations for the COVID-19 infection, as well as the observed time delay in epidemiological findings following public policy enactments. We use a compartment initialization logic coupled with a fitting technique to construct projections for key metrics to guide public health policy, including exposures, infections, hospitalizations, and deaths under a variety of social reopening scenarios. Our approach makes use of X-factor fitting and backcasting methods to construct meaningful and reliable models with minimal available data in order to provide timely policy guidance in the early phases of a pandemic.

## Introduction

Since its documented onset in December 2019 and formal identification in January 2020 in Wuhan, China, COVID-19 (SARS-CoV-2) has spread around the globe, infecting more than 7.5 million people globally by mid June 2020 [1]. In an atmosphere of intense uncertainty

infectious-disease-epidemiology/covid-19/
dashboards/index.php All additional relevant data
are within the manuscript with citations to external
sources.

**Funding:** The authors received no specific funding
for this work.

**Competing interests:** The authors have declared
that no competing interests exist.

around many of the epidemiological parameters for modeling including true case counts as a
result of low testing availability, the Modeling Emerging Threats for Arizona (METAz) Work-
group of Arizona State University has developed and refined models for predicting the burden
of disease in order to inform policy related to nonpharmaceutical interventions (i.e., social dis-
tancing). Because the burden of disease and transmission dynamics differ by location due to a
variety of factors including geography, population, and environmental conditions, METAz
chose to focus on state-level modeling to inform public health response efforts with greater
precision. The modeling approaches we describe can be applied to any region or state where
region-specific data are available. Here, we focus on the state of Arizona in the American
Southwest (population of around 7.3 million, 113,990 sq. miles, majority population concen-
trated in centrally-located Maricopa County) as a proof of concept.

Arizona's governor declared a state of emergency on March 11, and municipal govern-
ments began to enact limits on in-person gatherings and some business closures on March 16-
17. From March 30 to May 15, Arizona was under a stay-at-home order issued by the gover-
nor. As of June 10, Arizona has reported 29,852 cases across the state, with the majority of
cases in centrally-located Maricopa County, which includes 60 percent of the state's popula-
tion. Twenty-seven percent of the total cases were recorded in the first ten days of June. As of
June 10, Arizona's healthcare system has not experienced an overwhelming surge of COVID-
19 cases exceeding systemwide capacity to care for critically ill patients. Hospital admissions
appeared to have slowed and plateaued in April and May, indicating that social distancing
motivated by state and municipal policies enacted beginning in mid-March had reduced trans-
mission and may have been flattening the curve effectively in order to allow time to prepare
operations for future management of the disease in Arizona and avoid overwhelming hospital
systems as other states experienced.

However, as of early June, Arizona is experiencing increasing widespread community
transmission of SARS-CoV-2. Due to a relatively low rate of testing statewide, there is ongoing
debate and uncertainty about whether Arizona's case prevalence data provides an accurate
portrait of the true public health risk burden and whether we have passed an (initial) peak of
infections and hospitalizations statewide and in individual counties. Projections from a variety
of modeling groups (i.e., IHME, UA, ASU) had indicated that the peak number of cases will be
reached in Arizona in mid-April to mid-May. However, it is important to note that modeling
projections are inherently uncertain, and accurate assessment of case peaks will be possible
only once the peak has passed. In light of the transmission dynamics and laboratory reporting
delays for the SARS-CoV-2 outbreak, peak determination will be possible approximately two
to four weeks following peak occurrence. It is also important to note that there is still signifi-
cant uncertainty about the transmission dynamics of the virus, including the degree of asymp-
tomatic infection and transmission and the results do not capture the full range of uncertainty.
We demonstrate this observation through our modeling below.

On April 16, the United States Government released Guidelines for Opening Up America
Again, proposing a phased approach to reopening the country. In order to progress into and
through three sequential phases of opening businesses and other public and private services,
states are expected to meet a set of gating criteria outcome metrics along with a set of capacity
responsibilities for carrying out core public health and management functions. In order to
move to Phase 1 with limited reopening of businesses and other services, states must demon-
strate flattening the case rates, and in order to move to Phase 2 with expanded reopening of
businesses and services, states must demonstrate no rebound in case counts from the limited
reopening in Phase 1.

On May 15, Arizona's stay-at-home order expired, with targeted business openings
occurring on May 8 and May 11. At the time of reopening, Arizona had not met the CDC

gating criteria to move to Phase 1, nor had the state developed a comprehensive plan that incorporated the full testing capabilities within the state (both molecular and serological) with a program linked to non-pharmaceutical interventions (NPI) including stay-at-home and other social distancing and infection mitigation policies and procedures. In order to reopen Arizona safely, a phased approach needs to be data-driven and focused on avoiding a rapid surge in cases through appropriate and effective policy for non-pharmaceutical interventions.

Rising case counts and hospitalizations in late May and early June reflect that the move to lift policies restricting in-person interactions and the lack of statewide policies to enforce NPIs including physical distancing, masking, and hand hygiene resulted in markedly increased community transmission. As of June 10, there is not a statewide plan articulated to guide resumption of NPIs despite strong evidence of increased community transmission.

This paper proposes a mathematical framework that ties disease surveillance with the future burden on Arizona's hospital system and hospital resources to guide policy decisions in the early phase of a pandemic before extensive localized data are available. The mathematical model links together policy interventions with estimated outcomes for infections, hospitalizations, and deaths in an epidemiological analysis. One of the key features of our modeling methodology is the time-delay of new infections on confirmed case counts and the impact on the healthcare system. We propose methods to evaluate the likely outcomes for a range of policy decisions intended to keep Arizona safe while reopening in a responsible and defensible sequence.

# 1 Methods

## 1.1 Data sources

We use two publicly available data sources to initialize and fit our model: cumulative case counts and deaths in the State of Arizona between the dates of March 4 and June 7. Figs 1 and 2 depict the data that are used to obtain the results presented below. Both of these are publicly available and daily announced at the Arizona Department of Health Services' (ADHS) data dashboard at https://www.azdhs.gov/preparedness/epidemiology-disease-control/infectious-disease-epidemiology/covid-19/dashboards/index.php.

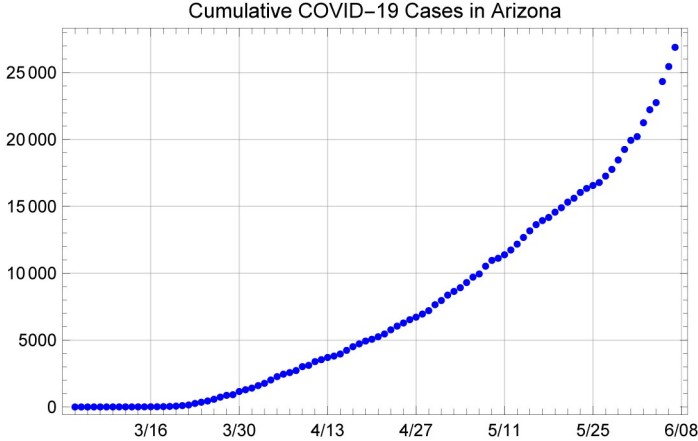

**Fig 1. Cumulative confirmed COVID-19 cases in Arizona, between March 4 to June 7, 2020.**

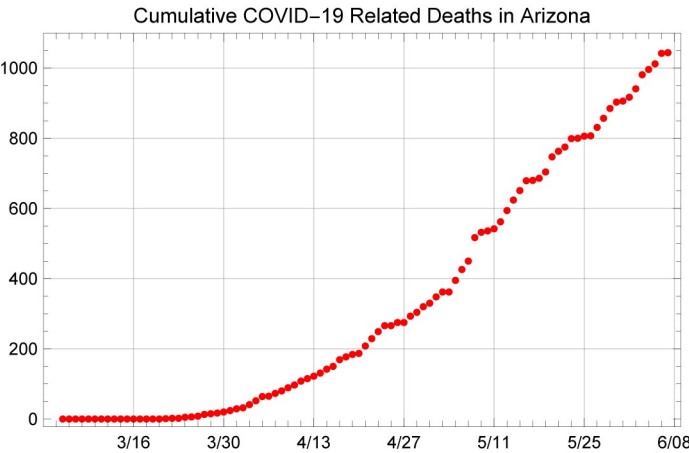

**Fig 2. Cumulative COVID-19 related deaths in Arizona, between March 4 to June 7, 2020.**

## 1.2 Structure of the model

We make use of a compartmental system dynamics model using a SEIRD framework that includes multiple compartments for infected individuals. This model structure allows us to estimate the number of patients in the hospital and assess model fit with respect to two sources of data: daily new cases obtained from the daily cumulative confirmed cases and daily cumulative reported deaths given in Figs 1 and 2. In essence, the population of interest, in this case, the population of the State of Arizona (assumed to be 7,278,717 in this study) is divided into states of Susceptible ($\mathcal{S}$), Exposed but not yet infectious ($\mathcal{E}$), Asymptomatic infected ($\mathcal{I}_a$), infectious and presymptomatic ($\mathcal{I}_p$), Symptomatic with a mild infection ($\mathcal{I}_s$), symptomatic with a severe infection and hospitalized ($\mathcal{H}$), symptomatic with a critical infection and in the ICU ($\mathcal{C}$), undergoing additional recovery in ICU ($\mathcal{B}$), Recovered and immune ($\mathcal{R}$) and Dead ($\mathcal{D}$), as shown in Fig 3.

Our model defines separate compartments for asymptomatic and presymptomatic individuals to explicitly account for differing rates of transmission and differing durations of illness. Individuals who are exposed to the virus go through a latent period (modeled by a rate of $\zeta$)

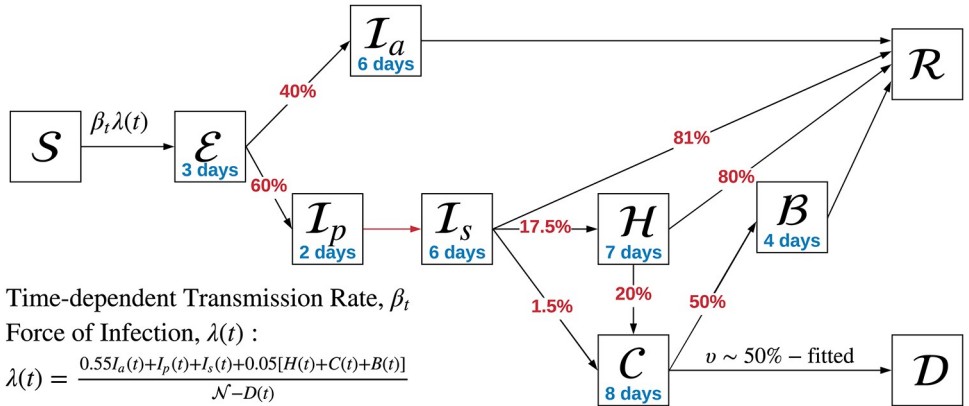

Time-dependent Transmission Rate, $\beta_t$
Force of Infection, $\lambda(t)$ :

$$\lambda(t) = \frac{0.55I_a(t) + I_p(t) + I_s(t) + 0.05[H(t) + C(t) + B(t)]}{\mathcal{N} - D(t)}$$

**Fig 3. Depiction of the compartmentalized system dynamics model used to represent transmission and disease progression for State of Arizona projections.**

during which they are exposed but not yet infectious. Following this latent period, we assume that some infections will recover without developing any symptoms (i.e., $\mathcal{I}_a$). The remaining infections will be presymptomatic for a period of time (modeled by a rate of $\delta$) meaning that their symptoms will remain subclinical, but they are assumed to be infectious during this time [2]. For symptomatic infections, the incubation time is taken to be 5 days, based on estimates by [3–7].

Due to the fact that our model includes multiple compartments with different characteristics of symptom onset and duration of infection, it is difficult to directly apply the serial interval/generation time estimates from the literature to estimate the duration of the latent period given that the classic definition of latent period depends on stability of the time to symptom onset. Therefore, for the novel coronavirus with variation in symptom onset, we use an averaged infectious period to estimate the latent period from estimates of serial interval. Recall that an approximation for the average latent period can be obtained by subtracting half the average infectious period from the average serial interval. Before explaining our assumption on the duration of infectiousness, we note that we used 6.6 days as the serial interval, which was supported by the early estimates available in the literature at the time of our study [4, 6, 8, 9] and was relevant to the local context that we were trying to model. Subsequent estimates of serial interval were slightly lower (around 5.5 days) due to faster identification of cases and contact tracing efforts [7, 10–13].

We assume that asymptomatic and symptomatic individuals are no longer infectious after they recover. The duration to recovery or death (i.e., the period of infectiousness) depends on the severity of the disease. We modeled a number of observed variations on how symptomatic individuals experience COVID-19. After the presymptomatic period, a large fraction, 81% (denoted by $\rho$ in the model) estimated by [14, 15], of symptomatic individuals go through a 6-day infectiousness period with relatively mild symptoms that do not require hospitalization. Similarly, we assumed that asymptomatic patients recover at a rate of $\gamma = 1/6$, corresponding to an average recovery duration of 6 days after the preinfectious period. This assumption was based on the estimates of infectious periods for mild infections available at the time of the study, which was somewhat scarce, since most of the available data was on severe or critical patients getting care at the hospital. We used [16] to justify an "average" duration of infectiousness of around 7 days for these patients, which subsequently proved to be a good assumption as shown by [17]. The remaining 19% of symptomatic patients develop a severe or critical infection and seek care at a hospital. We assume that these patients are generally isolated and infect others at a much reduced rate. We discuss this assumption further below. Using an average duration of infectiousness calculated as 0.4*6 + 0.6*8 = 7.2 days (using a weighted average of durations of infectiousness for asymptomatic and symptomatic patients), we can approximate average latent period as 6.6-3.6 = 3.0 days. Given the 5 day estimate for the incubation period, this gives us 2 days for the presymptomatic duration (modeled with rate $\delta$).

According to the available peer-reviewed literature, a large portion of the patients admitted to the hospital have a severe, but not critical, infection and recover after an average duration of 7 days in a regular hospital bed. An average of 20% of these patients, however, progress to a critical infection, requiring ICU care and possibly intubation. In addition to patients that progress to the ICU from a regular hospital bed, a small fraction of patients that present to the hospital with critical respiratory distress are directly admitted to the ICU. We have conferred with local clinicians in the Phoenix metropolitan area and Tucson (Arizona's major population centers) who confirm these patterns of patient progression through the hospital system. Hence, in our model, there are two modes of admission to the ICU; one directly from the emergency department and the other one from a regular ward, after the patient's infection progresses to a critical condition. The parameters for these splits are set to ensure that (i) the fraction of

symptomatic patients with mild infection is 81% [14, 15], (ii) the total fraction of symptomatic patients that develop a critical infection that requires ICU care is 5% [14] and (iii) 20% of patients in a regular bed progress to a critical infection [18].

Studies in the literature cite a diverse range of outcomes for patients in the ICU, but most agree that the ICU duration for patients that eventually recover is generally longer. For example, one study [19] cites point estimates for the duration of onset-of-symptoms to death to be 17.8 days and from onset-of-symptoms to hospital discharge to be 22.6 days. The additional time to discharge is due partly to various steps that caregivers have to take to arrange for care after the ICU period since generally patients that underwent intubation and other invasive procedures require subsequent care in other post-acute facilities. The additional post-acute recovery time is represented as another compartment, with a duration of 4 days (modeled with rate $\alpha$). The reported average ICU stays in the literature are generally very diverse; we adopted a conservative point estimate of 8 days to align with the symptom onset to recovery/death estimates [19] as well as other more detailed studies that tracked patients' progress through the hospital [15].

One of the important parameters in the model is $\omega$, which represents the fraction of asymptomatic patients. Several studies point to the importance of modeling transmissions by asymptomatic individuals, who may never be aware that they were transmitting the virus. However, point estimates on the fraction of individuals that experience asymptomatic infection vary greatly from context to context.

In our models we adopted an asymptomatic rate of 40% based on point estimates observed in multiple peer-reviewed manuscripts from different COVID-19 populations around the world [20, 21]. This assumption allows us to obtain worst-case estimates on the prevalence of infections in the general population given that, in the absence of widespread testing of asymptomatic individuals, the asymptomatic patients are generally undetected.

In our modeling and analysis, we explicitly consider the possibility that only a small fraction of the true incidence of infections are detected as COVID-19 cases and reflected in the reported case counts and deaths. One such example that points to a large undetected fraction of cases is [22], indicating that 86% of the early infections in China were undocumented, or in other words the "actual" cases in a population may be more than 7 times the detected cases. The same study also offers a rate of transmissions by asymptomatic individuals at 55% of the transmission rate by symptomatic individuals, which we reflect in the force of infection, $\lambda(t)$ shown in Fig 3. Subsequently several other papers have offered additional understanding on the role of asymptomatic infections in transmission and its prevalence in different contexts [2, 20, 21, 23–25]. We use these papers along with the actual data on new cases and deaths in Arizona to obtain point estimates for model parameters. We also devise an initialization algorithm to identify initial values of the compartments in the model.

The ordinary differential equations (ODE) that define the system dynamics are given by Eqs (1) through (10).

$$\mathcal{S}'(t) \quad = \quad -\beta_t \, \lambda(t) \, \mathcal{S}(t) \tag{1}$$

$$\mathcal{E}'(t) \quad = \quad \beta_t \, \lambda(t) \, \mathcal{S}(t) - \zeta \, \mathcal{E}(t) \tag{2}$$

$$\mathcal{I}'_a(t) \quad = \quad \zeta \, \omega \, \mathcal{E}(t) - \gamma \mathcal{I}_a(t) \tag{3}$$

$$\mathcal{I}'_p(t) \quad = \quad \zeta \, (1 - \omega) \, \mathcal{E}(t) - \delta \, \mathcal{I}_p(t) \tag{4}$$

$$\mathcal{I}'_{s}(t) \;=\; \delta\,\mathcal{I}_{p}(t) - \gamma\,\mathcal{I}_{s}(t) \tag{5}$$

$$\mathcal{H}'(t) \;=\; \gamma\,\psi\,\mathcal{I}_{s}(t) - \mu\,\mathcal{H}(t) \tag{6}$$

$$\mathcal{C}'(t) \;=\; \gamma\,(1 - \rho - \psi)\,\mathcal{I}_{s}(t) - v\,\mathcal{C}(t) + \mu\,\phi\,\mathcal{H}(t) \tag{7}$$

$$\mathcal{B}'(t) \;=\; v\,(1 - \upsilon)\,\mathcal{C}(t) - \alpha\,\mathcal{B}(t) \tag{8}$$

$$\mathcal{R}'(t) \;=\; \gamma\,\mathcal{I}_{a}(t) + \gamma\,\rho\,\mathcal{I}_{s}(t) + \mu\,(1 - \phi)\,\mathcal{H}(t) + \alpha\,\mathcal{B}(t) \tag{9}$$

$$\mathcal{D}'(t) \;=\; v\,\upsilon\,\mathcal{C}(t) \tag{10}$$

We assume a time dependent force of infection equal to $\beta_t \lambda(t)$, where

$$\lambda(t) = \frac{0.55 I_a(t) + I_p(t) + I_s(t) + 0.05[H(t) + C(t) + B(t)]}{\mathcal{N} - D(t)} \;. \tag{11}$$

The $\lambda(t)$ term can be thought of as the probability that an arbitrary individual is infectious at a given time, $t$.

This expression is motivated by the fact that asymptomatic individuals transmit the disease at a reduced rate as discussed above, and COVID patients with severe or critical infections who are typically receiving care in the hospital (i.e., compartments $\mathcal{H}$, $\mathcal{C}$ and $\mathcal{B}$) are relatively well isolated via institutional infection control measures. Hence, they only transmit at a rate that is equal to 5% of the presymptomatic or symptomatic patients.

Studies that point to the high infectiousness of presymptomatic patients [27, 28] imply that infections are mostly driven by patients in these compartments. Hence, we assume that presymptomatic and symptomatic patients transmit the disease at the same rate.

We model a time-dependent transmission rate, $\beta_t$, denoted by the subscript $t$ to represent the time dependency. This term represents the average rate of contact between susceptible and (symptomatic-equivalent) infectious people multiplied by the probability of transmission given contact. The rate at which individuals become exposed to the virus at time $t$ is strongly driven by the term, $\beta_t$. A good way of thinking about the impact of non-pharmaceutical interventions such as social distancing, stay-at-home orders, school closures, wearing masks, etc. is through the term $\beta_t$, and how the different interventions impact either (i) the average number of infectious individuals that susceptible individuals contact, or (ii) the probability of transmission given contact.

Note that an increase in either of these two values would lead to an increase in the effective transmission rate at a given time, which will then increase the rate at which susceptible individuals get exposed to the virus. Keeping the same overall transmission rate the same while increasing the average rate of contacts requires that the probability of transmission given contact be reduced from through measures that reduce the probability of transmission given contact with an infectious individual. Such measures may involve hand washing practices, wearing masks, keeping 6+ ft apart, etc. As the interactions between individuals are expected to increase after the stay-at-home orders are lifted, the importance of such measures should be more rigorously emphasized.

The model parameters and point estimates for them obtained from the literature are provided in Table 1. Our approach of initializing the compartments and fitting the transmission

**Table 1. Point estimates used for model parameters and sources.**

| Description | Parameter | Value | Sources |
|---|---|---|---|
| Time to infectiousness | $\zeta^{-1}$ | 3 days | [4, 6, 8, 9] |
| Presymptomatic duration | $\delta^{-1}$ | 2 days | [3–7]. |
| Asymptomatic infectious period | $\gamma^{-1}$ | 6 days | [16, 17] |
| Mild infection recovery time | $\gamma^{-1}$ | 6 days | [16, 17] |
| Severe infection recovery time | $\mu^{-1}$ | 7 days | [14, 26] |
| Critical infection to death | $v^{-1}$ | 8 days | [15, 19] |
| Additional days to recover after ICU | $\alpha^{-1}$ | 4 days | [19] |
| Fraction of asymptomatic infections | $\omega$ | 40% | [20, 21] |
| Fraction of mild symptomatic infections | $\rho$ | 81% | [14] |
| Fraction hospitalized on regular bed | $\psi$ | 17.5% | [14] |
| Fraction of hospitalized progressing to ICU | $\phi$ | 20% | [18] |
| Mortality among ICU patients | $v$ | 50-60% | data fit |

rate $\beta_t$ and the mortality rate at the ICU, $v$, uses publicly available data on case counts and COVID-19 related deaths in Arizona. In our fitting procedure, we allow for the transmission rate, $\beta_t$ to change in response to significant events or policy changes, such as non-pharmaceutical interventions being enacted or lifted in Arizona, as we explain below.

## 1.3 Initialization of compartments

We first present a methodology to initialize the model in a manner that is independent of the transmission rate, $\beta$. In particular, we consider the data on the cumulative number of confirmed cases in Arizona, where the first reported cases of community transmission were on March 4, as shown in Fig 1. We use these data to obtain the number of new cases on each day. The average reporting delay on COVID-19 is about 6 days in Arizona. Given that our model indicates an incubation period of 5 days and average time to seek testing (when it is available) is about 3 days after symptom-onset, we obtain presumed exposure dates for the reported new cases on each day (i.e., 14 days before a case is confirmed). A visual that shows this logic is shown with the blue bars (reported new cases over time) and the orange bars (numbers eventually detected, shown on the presumed exposure dates) in Fig 4. Note that the orange bars show the number of individuals exposed to the virus on the given day, who are then eventually detected by testing.

As discussed above, a large portion of the individuals exposed to the virus on a given day are never detected due to the fact that (i) a significant portion of these individuals never develop symptoms; and (ii) some symptomatic individuals are never tested, their infections are attributed to another influenza-like illness, or their case is missed due to false negative results in COVID-19 tests. To account for the large rate of undetected infections, we have devised an intuitive approach using an X-factor initialization scheme where we multiply the number of eventually detected-exposed individuals by the X-factor to estimate the underlying overall exposures on a given presumed exposure day. Note that this is a relatively crude method of obtaining an "average" number of exposures on each day, ignoring the large number of uncertainties that are involved in the observations of daily case counts. As a result, we note that the prediction intervals we note below may somewhat be underestimated. In subsequent work, we have developed methods that recognize that the number of case counts resulting from a given number of exposures would be random (rather than a deterministic relationship like the one we are using here). In particular, we randomly generate exposures on randomly generated

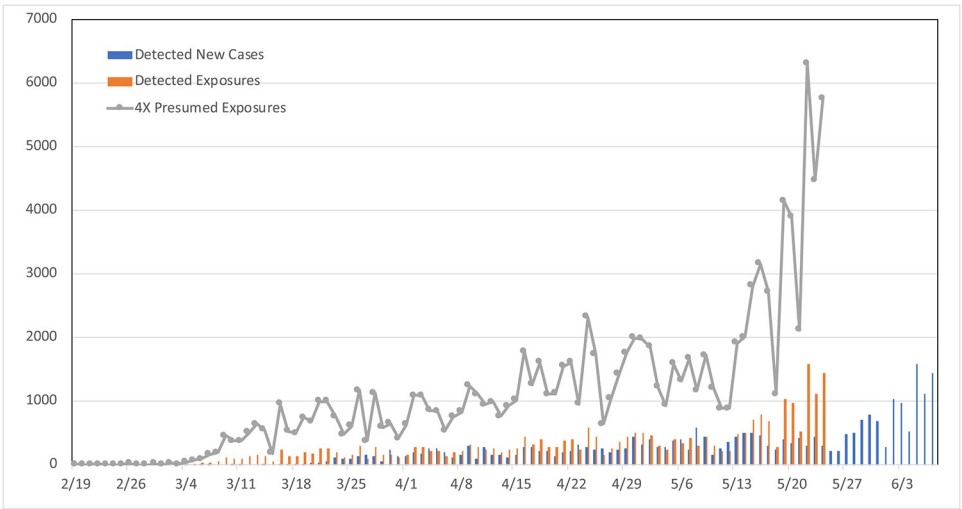

**Fig 4. Reported new cases and presumed exposure dates.**

exposure dates based on observed data, but we have observed that such approaches did not make a significant impact on the performance of the initialization algorithm.

The X-factor determination in this scheme is highly correlated to the degree to which the testing procedures are able to detect the infections in the system. Given our assumption that 40% of all infections are asymptomatic, the minimum X-factor that is aligned with our modeling assumptions is 1.67, since these individuals are almost never tested and confirmed due to the fact that they do not exhibit symptoms to prompt testing. At the upper end, our model indicates that about 12% of infections have severe or critical infections, requiring them to seek healthcare. This implies that the maximum X-factor that would be aligned with our model is about 8, since nearly all individuals seeking care in Arizona for COVID-like symptoms are tested for COVID-19. In Fig 4, the grey points depict exposures in a scenario using X-factor of 4.

The X-factored exposures on presumed exposure days are then fed into our SEIRD model, keeping the transmission rate to zero. We obtain an approximate continuous time function by interpolating over these presumed exposures, called $W(s)$. Fig 5 shows the approximated rate of exposures over time in X-factor of 4 scenario between March 4 and March 29 in Arizona; the black dots are the daily presumed exposures also shown in Fig 4.

We then numerically evaluate the convolution

$$E[N_i(t)|W(s), 0 \leq s \leq t] = \int_0^t W(s)f_i(t-s)ds \tag{12}$$

to obtain the expected number in compartment $i \in \{\mathcal{I}_a, \mathcal{I}_p, \mathcal{I}_s, \mathcal{H}, \mathcal{C}, \mathcal{B}, \mathcal{R}, \mathcal{D}\}$ at time $t$, where $f_i(\tau)$ denotes the probability that an individual would be in compartment $i$ $\tau$ time units after exposure to the virus. The $f_i(\cdot)$ functions for each compartment in the model can be obtained by simulating the above stated model with one exposed individual and transmission rate of zero. Alternatively, we could have discretized time and calculated the estimated number in each bin at a desired time, given the presumed exposures in each day prior to that point. For example, the process of estimating the number in the symptomatic compartment on a given day $t$ would involve taking the presumed exposures in each day prior to that date, calculating the expected number that will be in the symptomatic compartment by day $t$ out of those

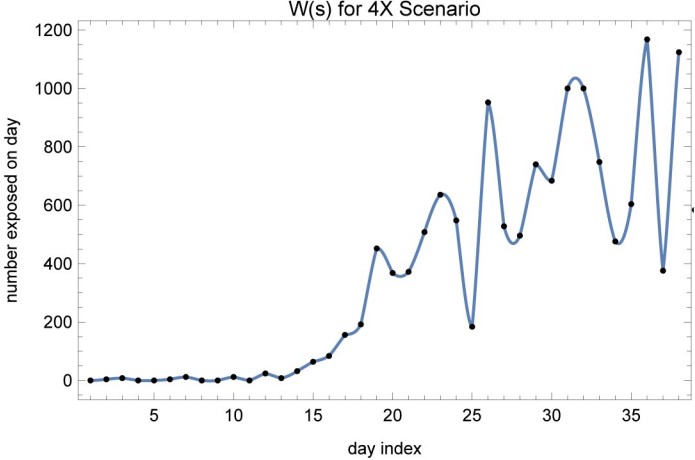

**Fig 5. The $W(s)$ function for the X-factor of 4 scenario, obtained by inflating the daily new cases.**

presumed exposures, and summing over all of the days prior to date $t$. The convolution essentially performs this calculation on a continuous time basis. As an example, Fig 6 shows the fraction at the hospital, $f_{\mathcal{H}+\mathcal{C}+\mathcal{B}}(\cdot)$ versus time.

The solution to the ODEs is unique given a set of initial values for the number in each compartment at time zero. Using the above initialization logic, we calculate the number that we expect to see in each compartment on a chosen presumed exposure day, using all of the data on new cases reported on the presumed exposure days prior to this point, and using the number of presumed exposures on that day to initialize the $\mathcal{E}$ compartment. We are then -almost-ready to simulate the model starting from that day and observe the number in each compartment to obtain projections.

### 1.4 Fitting transmission rate and mortality

In our study, we initialize the compartments on March 30 (i.e., this calendar day is our $t = 0$) and use the actual data on presumed exposures (under any assumed X-factor scenario) starting

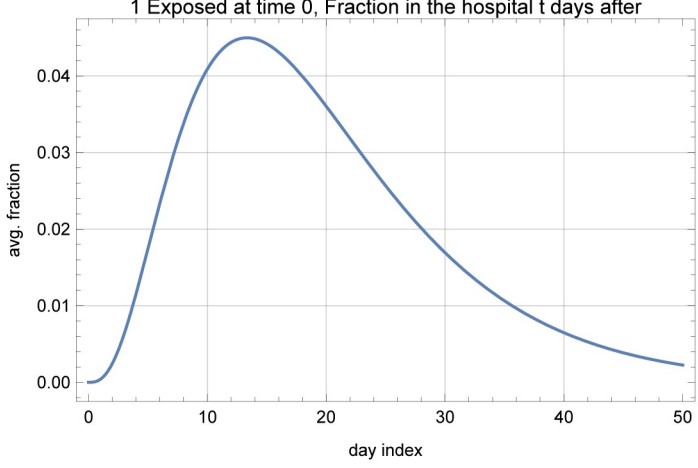

**Fig 6. $f_{\mathcal{H}+\mathcal{C}+\mathcal{B}}(t)$ under assumed parameters.**

from March 31 until May 24 to fit the transmission rate, $\beta_t$. Note that the presumed exposure date of May 24 corresponds to the actual reporting date of June 7, which is the last data point that we use in our results in this manuscript. We tried a number of different initialization dates, and the results on the transmission rate fit were comparable. We divided the time between March 31 and May 24 for which we have presumed exposures data into three periods correlating to dates of significant changes in Arizona public policy and activities related to NPIs including business closures and stay-at-home orders. We fitted three possibly different $\beta$ values to each period, resulting in a piece-wise constant transmission rate structure. In particular, we assumed constant $\beta$ values between March 31 to April 15 (which we refer to as $\hat{\beta}_1$, representing early adjustment to the stay-at-home order enacted on March 30), April 16 to May 10 (which we refer to as $\hat{\beta}_2$, representing stabilization of public response to the stay-at-home order) and May 11 through May 24 (which we refer to as $\hat{\beta}_3$, representing reopening of some businesses and activities including personal care services on May 8, dine-in restaurants on May 11, and the expiration of the stay-at-home order on May 15). We assume that $\hat{\beta}_3$ is the best available transmission rate estimate explaining the exposures beyond May 24 (since at the writing of this manuscript no changes in the non-pharmaceurical interventions have been announced) and use that value to generate the projections below for exposure dates later than May 24.

We use Wolfram Mathematica 12 to obtain a numerical solution to the ODEs and obtain a parametric function that describes the number of susceptibles in the system given the initial population of 7,278,717 (population of Arizona) and the assumed loading scenario described by the X-factor used when initializing the compartments. We then use the X-factored presumed exposures to obtain the corresponding number of susceptibles, which is equal to the initial number of susceptibles in the population minus the cumulative number of exposed individuals, and use a nonlinear model fit procedure to estimate $\hat{\beta}_1$, $\hat{\beta}_2$ and $\hat{\beta}_3$. As an example, Fig 7 shows the model fit along with the 95% prediction intervals under a 4X scenario. In addition, we plot the model predicted and 4X presumed exposures in Fig 8.

In addition to fitting the transmission rate, $\beta$, we use the cumulative number of COVID-19 related deaths in Arizona to fit the mortality rate among the ICU patients. Note that in our model, we assume that all patients who die will do so in the ICU, which ignores the deaths that occur outside the hospital. At the time of the writing of this manuscript, Arizona's healthcare capacity, beds, and ICU have been sufficient to care for COVID-19 patients.

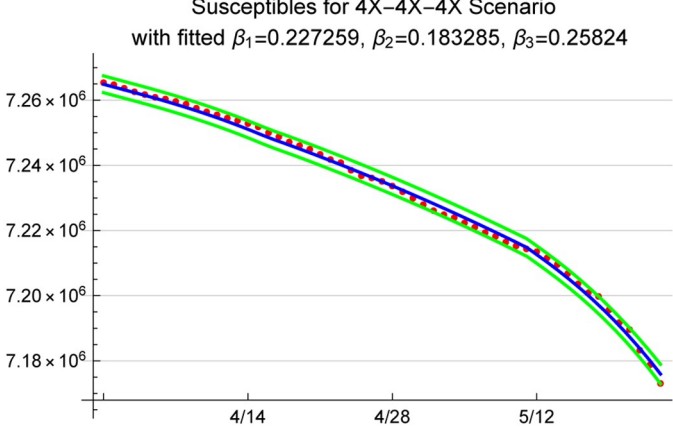

**Fig 7. 95% prediction bands for susceptibles; red dots show presumed susceptibles under 4X scenario.**

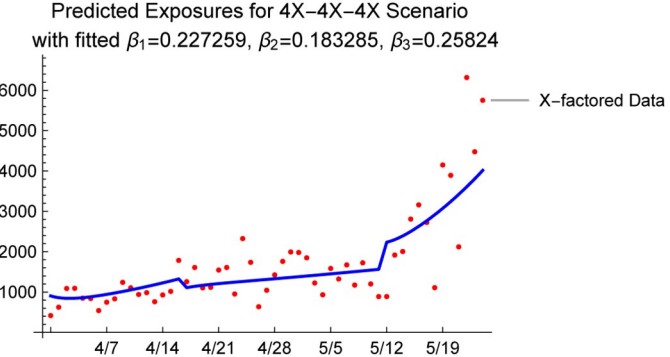

**Fig 8. 4X presumed exposures from data, and predicted exposures with the fitted $\beta$ values under 4X.**

Therefore, to our knowledge, Arizona has not experienced significant reported deaths outside the hospitals due to an inability for patients to access critical care services. There is, however, ongoing debate about whether COVID-19 related deaths are under-reported in Arizona and nationwide.

Given our assumption for this modeling exercise that deaths are primarily occurring in the ICU, for this analysis we assumed that the information on the reported deaths is relatively accurate; hence, we do not amplify the reported deaths when fitting the death rate $v$. Fig 9 shows the cumulative number of deaths that the model predicts under a 4X loading scenario. Note that under the 1.67X loading scenario, the ICU death rate produced by the model fit procedure was on the order of 1.24. That is, the assumption of 88% detection rate was not aligned with the point estimates we used in the model to predict the reported death rates. Given that there is widespread belief that COVID-19 deaths are underreported, we understand this finding to be in support of the idea that only a fraction of the infections are detected, and thus reported in the official case counts. In the next section, we present projections for 1.67X, 4X, and 6X loading scenarios to provide a range of future projections for cases, hospitalizations, and deaths.

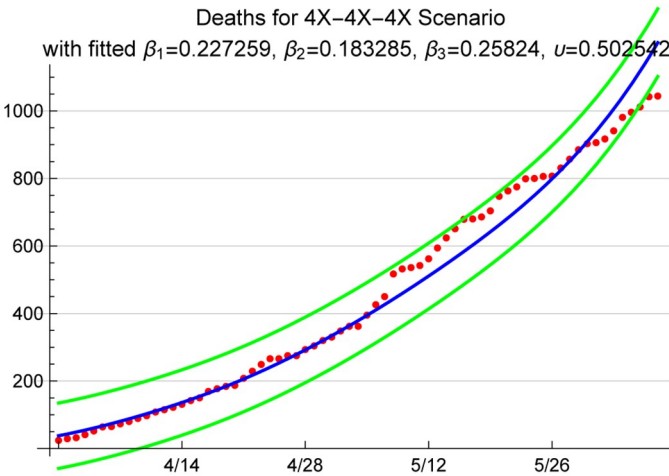

**Fig 9. Model predicted cumulative number of deaths between 3/31 and 6/7 with 95% prediction intervals; red dots are the reported COVID-19 deaths for the period in Arizona.**

## 2 Results

We provide projections on the number of deaths, number of people hospitalized, and total infections for a number of cases that differ in X-factor and the transmission rate over time. We first start with the benchmark cases of 1.67X, 4X and 6X loading scenarios simulated under the assumption that the transmission rate stays at the fitted $\hat{\beta}_3$ for exposure dates beyond May 24. It is useful to observe the dynamics for a relatively long horizon of 500 days, as given in Fig 10.

Day 1 in this simulation is March 31, where we initialize our model and run it with the $\beta$ and $\upsilon$ values that we fitted using the X-factored data on new cases and deaths after this point (i.e., 55 points of backcasted presumed exposures and 69 days of data on deaths). We define herd immunity as the point at which wide community spread is suppressed due to the large proportion of individuals in the population with immunity. Given the large initial susceptible population that we use for the model (i.e., 7,278,717) herd immunity is reached around late October at the currently fitted transmission rate. This figure demonstrates that policies that were initially enacted in March to limit close person-to-person contacts were effective in reducing transmission during April and early May, but also preserved a high pool of susceptibles in the general public to fuel future outbreaks under conditions where NPIs are not effectively implemented.

While visualization of the epidemiological curves is useful to gain insights into the long-term behavior and other concerns such as peaks and herd immunity, it is more informative to focus on shorter-term projections since it is unlikely that $\beta$ remains constant over a very long period of time due in real-world conditions due to fluctuations with regard to NPI measures taken by individuals and public health officials.

The baseline plots given in Fig 11 through Fig 14 show the total infected and hospitalizations as well as exposures and deaths under the 1.67X, 4X and 6X loading scenarios with fitted $\beta$ and $\upsilon$. The fitted values in each scenario are shown in the plot legends. Recall that to fit the mortality rate in each scenario, we kept the data on reported deaths intact and fitted the value of $\upsilon$ to the data. Note that the fitted $\upsilon$ value of 0.50 for the 4X case results in a mortality rate of 2.5% among symptomatic individuals. For the 1.67X scenario, this resulted in a fitted $\upsilon$ value of 1.23, meaning that the 1.67X scenario did not generate sufficient number of patients in the ICU to explain the reported deaths in Arizona. Hence, we used an $\upsilon$ value of 0.99 for the 1.67X runs. This result shows that 1.67X scenario, which essentially represents a case where all of the symptomatic individuals were detected by the testing effort, was overly optimistic and fell

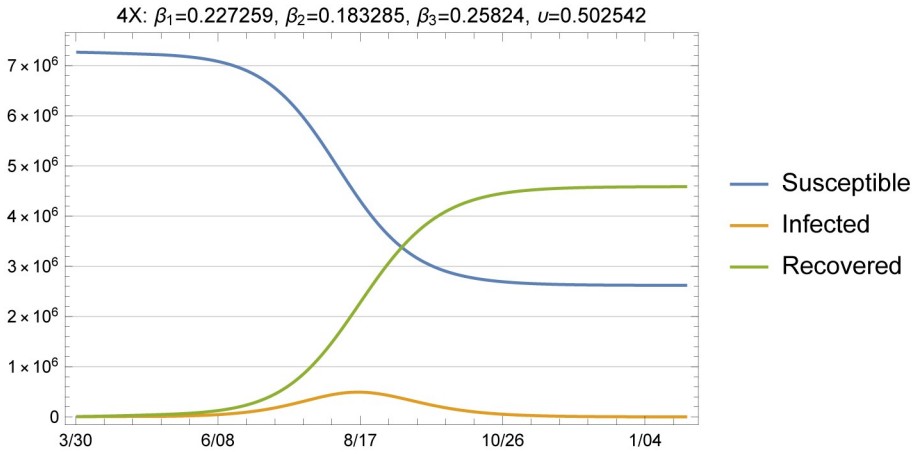

**Fig 10. Susceptible, infected and recovered, 4X loading with fitted $\beta$ and $\upsilon$.**

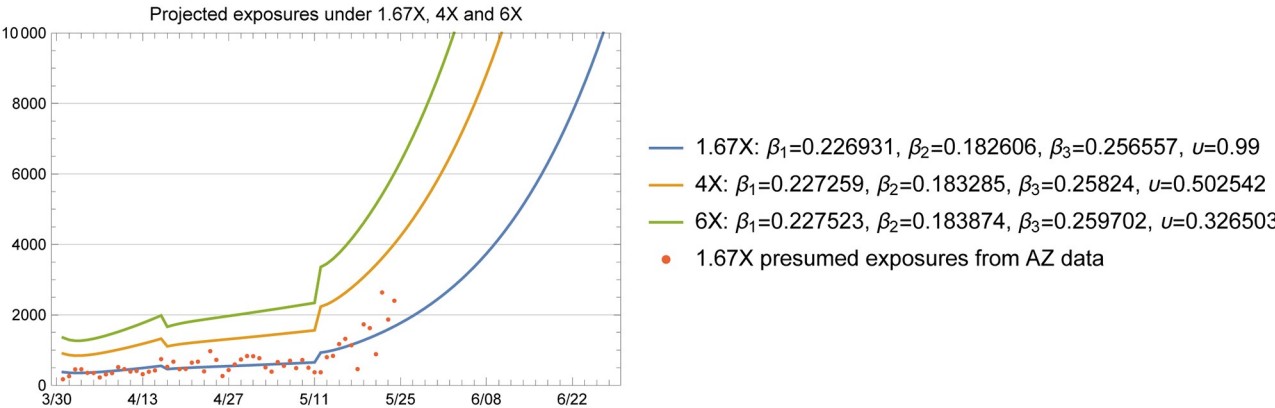

**Fig 11. 1.67X exposures inferred from actual data (red dots) and projected by the model.**

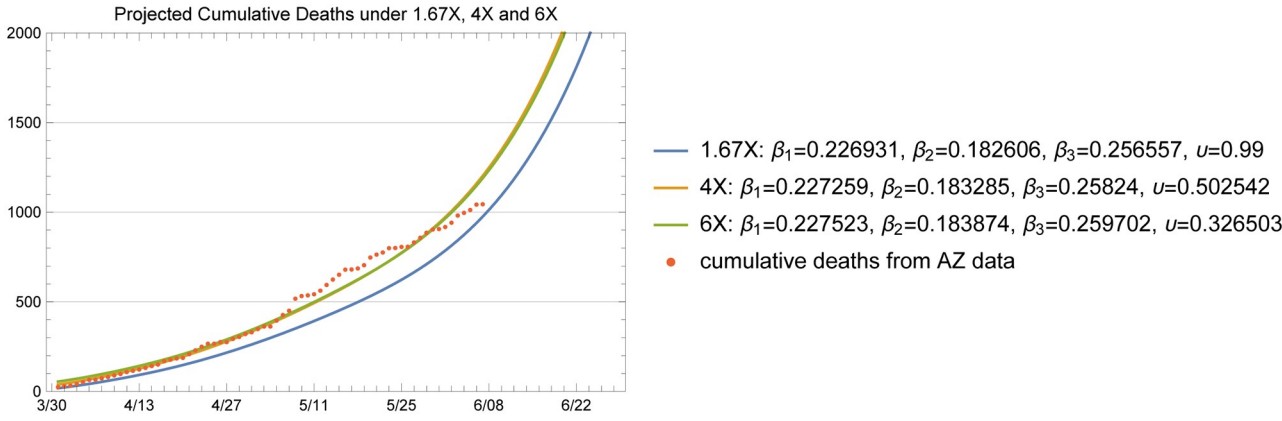

**Fig 12. Cumulative number of deaths; actual data (red dots) and projected by the model.**

severely short of explaining the observed deaths in Arizona over time. However, this case still serves a significant practical purpose, since it demonstrates that the actual X-factor in the system is higher than 1.67, and gives a lower bound on the projections that one may develop based on the available data. In fact, this case was highly useful in our conversations with the

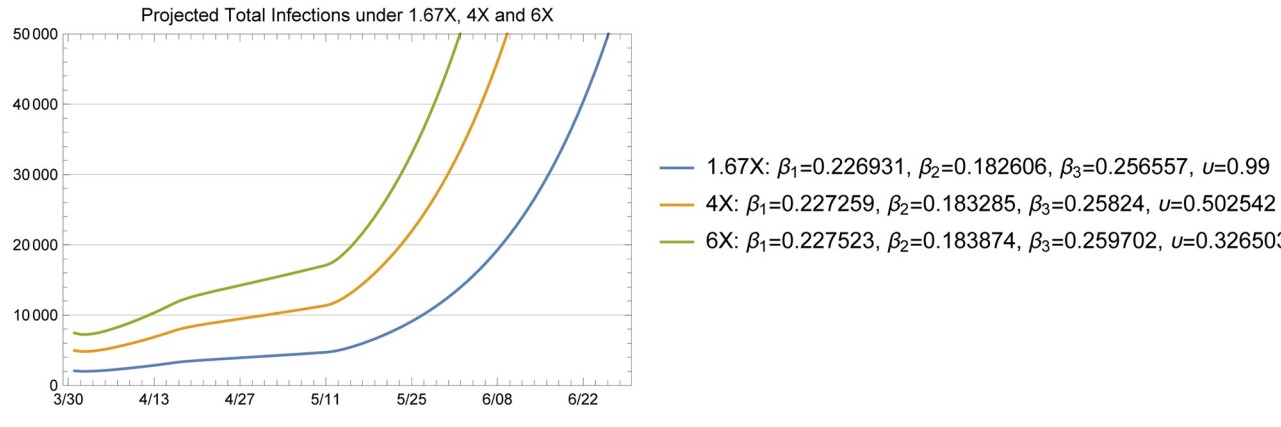

**Fig 13. Total infected projected by the model with fitted $\beta$ and $v$.**

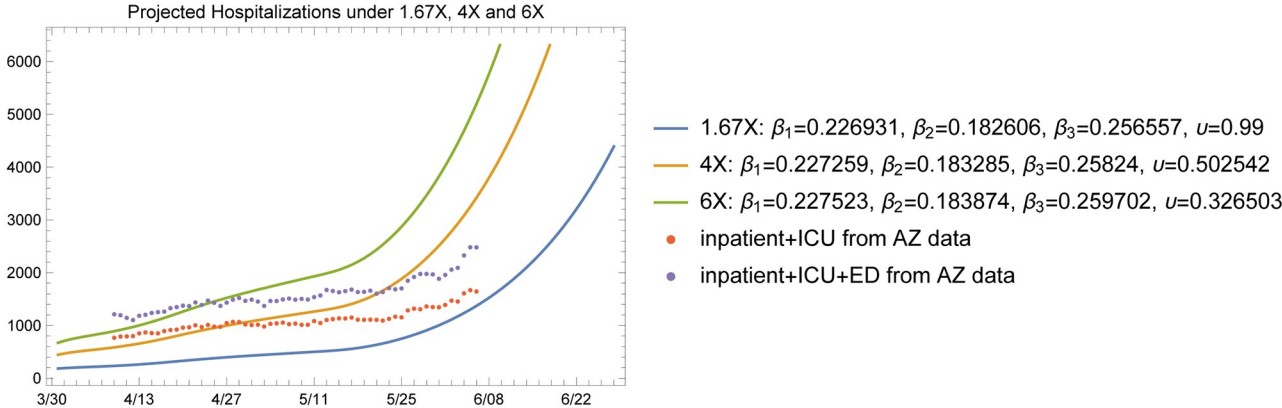

**Fig 14. Hospitalized patients projected by the model with fitted $\beta$ and $v$.**

public health experts in the State, to guide their testing efforts and reflect a reasonable picture of the epidemic during the early stages.

In comparison, the fitted upsilon for the 6X scenario was 0.33 since this scenario assumed a higher undetected rate and loaded more patients at the ICU. An alternative approach would have been to inflate the death numbers to account for the observation that deaths related to COVID-19 may be underreported. We do not use this approach in our projections in order to maintain an evidence-based conservative set of estimates on the death toll of the epidemic.

Fig 14 shows the projected hospitalizations (the sum of the numbers in compartments $\mathcal{H}$, $\mathcal{C}$, and $\mathcal{B}$) along with the available hospital data between April 9 and June 7. Again, these data are obtained from the ADHS data dashboard, where the census of inpatient, ICU, and emergency department (ED) bed usage are provided separately. Note that in our model, we do not have a separate ED compartment, so we plotted the hospital data from the ADHS website in two ways: one without the numbers from the ED and one with the numbers from the ED. We note that a significant portion of the patients in the ED on each day may be discharged and sent home to recover rather than being admitted to an inpatient bed. As shown in Fig 14 the projected hospitalizations fall within the 1.67X and 4X scenarios in both of our treatments of ED data.

The model projections show that it is reasonable to expect a slowly increasing number of patients in the hospital in the short term (i.e., late May to mid June) with subsequent rapid growth of hospitalization rates with increased community transmission as NPI policies were lifted on May 15. This increase in cases is due to the large number of susceptible individuals in the population resulting, in part, to the effectiveness of NPI policies in place from March 30 to May 15.

## 2.1 Projections under a favorable summer effect

At the time of the writing of this manuscript, there was significant debate about the impact of higher summer temperatures in large regions of Arizona on the transmission rate of COVID-19; there was no consensus in the peer-reviewed literature. This conversation about possible temperature effects was important since some local policymakers were raising the possibility that a heat-mediated summertime transmission decline could justify lifting NPI policies implemented during the relatively cooler spring months. A so-called summer effect would include a potentially suppressive effect on virus survivability in the extremely elevated temperatures and UV radiation of the desert Southwest. Simultaneously, the extreme summer heat in Arizona's

population centers also creates a behavioral effect, changing patterns of indoor and outdoor activity in Arizona's desert environment. In essence, Arizona's summer effect behaviorally mimics the winter effect in more temperate regions, as people seek heat relief in indoor environments. Baker et al.'s [28] published in May 2020 indicates that nonpharmaceutical control measures may moderate the pandemic-climate interaction through susceptible depletion. In other words, human behavior can dominate any climate effect at this early stage of the pandemic, which clearly deviates from the behavior observed for endemic infections.

Our experience in Arizona confirms the findings in this paper; the transmission rate observed from data was more shaped by the NPIs in effect than by the weather. At the timing of this study, however, given the uncertainty about a potential summer effect, particularly on virus survivability, we found the following analysis to useful to demonstrate the sensitivity of outbreak dynamics to the transmission rate $\beta$ when presenting our results to public health officials. The scenarios demonstrate the impact of different levels of favorable summer effects on projected hospitalizations.

In particular, we have simulated four scenarios with no summer effect, 25% decrease in transmission rate on 5/29, 25% transmission rate on 6/12, and a 50% decrease on 6/12 and calculated the projected hospitalizations in each case. The projections under these cases were useful for public health officials in the State to appreciate the fact that given all the debate with respect to a favorable summer effect, it was likely that hospital resources in Arizona during the summer of 2020 will be significantly constrained. These projections are provided here to demonstrate how we used such scenarios to communicate these issues, rather than an expectation that these scenarios are epidemiologically relevant or supported by data or studies on potential summer effects.

In Fig 15, we plotted the projected 1.67X hospitalizations under four different scenarios with respect to the summer effect. We plotted the projected hospitalizations for a longer horizon to show the impact of a favorable summer effect on transmission rates. We chose to use the 1.67X scenario for this purpose since the current hospitalization data seems to be closer to the 1.67X projections. The plot shows the tradeoff between an early summer effect versus a later but more significant summer effect. The figure also demonstrates the impact of a 25% to 50% reduction in the transmission rate as well as the impact of the timing of the summer effect in further flattening the curve.

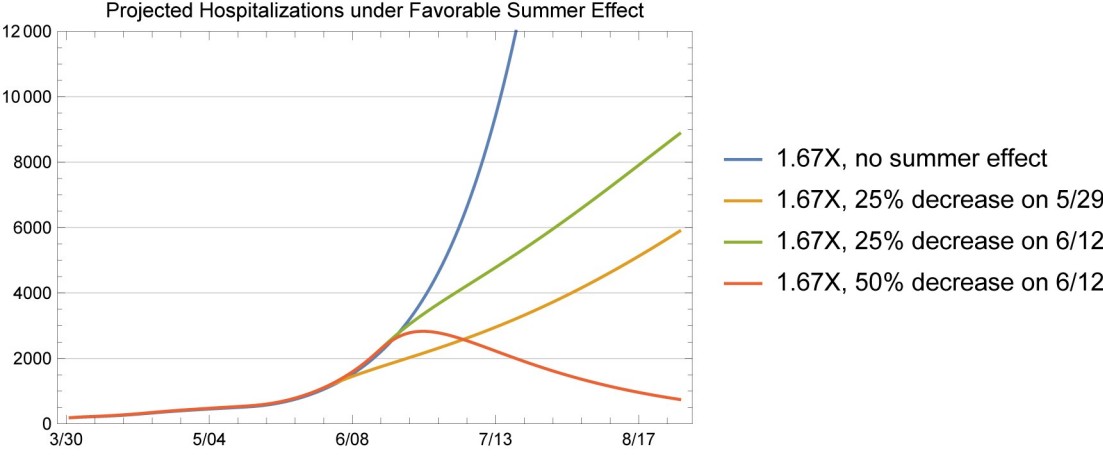

**Fig 15. Hospitalization projections under favorable summer effect scenarios for 1.67X.**

## 2.2 Initiating non-pharmaceutical interventions

The current estimation procedure applied to the current epidemiological data as of June 7 results in a significant increase in the transmission rate starting from May 11, which we use as the transmission rate estimate to obtain baseline projections as shown in Fig 11. Of note, May 11 was the date on which restaurants were allowed to resume dine-in operations statewide. This baseline reflects the observed data from the state health department reporting system at the present time during the response. We note that there are frequent data reporting corrections and so the current estimation targets could potentially change. The model includes multiple changes in transmission rates correlating with policy implementations. When NPIs were initiated, including bar closures and restaurant restrictions in the urban centers on March 17 and the statewide stay-at-home order on March 30, transmission rates were reduced. When NPI policies were lifted, including the resumption of dine-in restaurant service on May 11 and the end of the stay-at-home order on May 15, transmission rates increased.

It is worthwhile to consider the impact of initiating NPIs once more in an effort to return to the transmission rate estimated by our model during Arizona's stay-at-home order. We analyze the reduction in the number of infections and the number of hospitalized patients that may result from the reinitiation of NPIs at different points in time, starting from June 8. Recall that the latest data point we used in this analysis is June 7 data on cumulative confirmed COVID-19 cases and deaths, so June 8 represents the earliest point in time that the NPIs could be reinitiated. We do not specify any specific NPIs to achieve this reduction in transmission. Rather, we use the prior estimates in transmission parameter $\beta$ as a measure of lowered transmission under heightened policy implementation.

To represent the patient care load imposed on Arizona's hospitals, we consider the area under the total infections and hospitalizations curves, in a manner similar to the calculation of "illness inventory" or utilization in the system over time. We consider five dates that NPIs can be initiated: June 8, June 15, June 22, June 29, and July 6 and obtain the following improvement metric in comparison to the baseline case of no NPIs, which assumes that the transmission rate stays constant at the $\hat{\beta}_3$. The results are qualitatively similar, so to provide some conservative estimates, we use 1.67X for this analysis. Hence, we use $\hat{\beta}_1 = 0.226931$, $\hat{\beta}_2 = 0.182606$, $\hat{\beta}_3 = 0.256557$, $\hat{\upsilon} = 0.99$. At the indicated NPI initiation times, we revert the transition rate to the lowest under NPI, which is $\hat{\beta}_2 = 0.182606$.

First, it is useful to observe the total infections (including asymptomatic individuals, presymptomatic individuals, symptomatic individuals and hospitalized patients) under the baseline and the five intervention time options over the next several months under these assumptions to compare the behavior under these different scenarios. We note that we are not presenting this figure to provide projections, but rather provide a visual reference to explain the difference in long-run behavior that NPIs, through the reductions in transmission rate implies. In Fig 16 we see that the NPIs result in significantly different infection patterns as a result of the reduction in transmission rates. While this simulation represents a highly optimistic scenario with respect to the impact of NPI reinitiation on the viral transmission rate, this analysis clearly shows that initiation of NPIs can provide significant relief on the healthcare resource demands that the pandemic presents, even in the setting of elevated baseline infection rates with a large susceptible population.

We have indicated that this analysis presents an optimistic scenario with respect to the impact of the re-initiation of NPIs on the transmission rate due to several reasons. First, individual behavior with regard to NPIs including masking and physical distancing varies due to individual beliefs and adherence to recommended actions. Second, some population segments

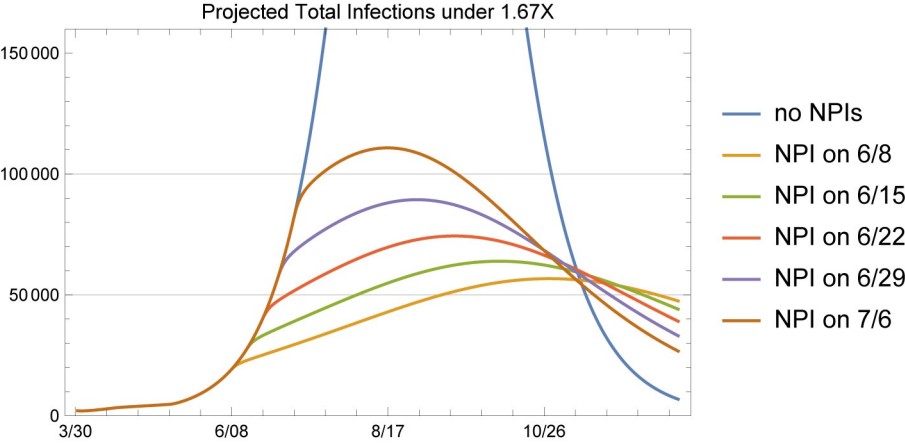

**Fig 16. Total infections under the baseline and five NPI initiation date options for 1.67X.**

experience systemic inequity including inadequate access to supply chain and economic resources to acquire protective equipment (e.g. face masks) and hygiene resources (e.g. hand sanitizer) to afford measures of protection. Third, individuals experiencing systemic resource inequities are least able to avoid public contact and practice physical distancing due to fragile employment status in front-line jobs. Fourth, the high asymptomatic rate and presymptomatic transmission patterns coupled with low viral testing rates and extremely limited contact tracing capacity statewide has limited the ability of the public health system to contain outbreaks even in the setting of optimal NPI adherence by individuals in the community.

To quantify the improvement that can be expected from the re-initiation of NPIs on a given date, we consider the following percent reduction metric, which compares the areas under each curve over time. That is,

$$\rho_m^k(\tau) = 1 \ - \ \frac{\int_0^\tau N_m^k(t)dt}{\int_0^\tau N_m(t)dt} \ , \tag{13}$$

where the superscript $k$ denotes the NPI initiation date options and superscript $m$ can be total infections, hospitalized patients or deaths. We calculate the reduction metrics for 1.67X however, the results are similar qualitatively under 4X and 6X.

Table 2, which provides the percent reductions in total infections, shows that initiating NPIs results in significant reductions in total infections. For example, initiating NPIs on June 22 would imply a reduction of 72% in the total infections in Arizona by the beginning of

**Table 2. Percent reductions observed in the total infections by the indicated dates for each NPI initiation date option.**

| | Percent reduction in total infections with NPI initiations | | | | |
|---|---|---|---|---|---|
| by date | NPI on 6/8 | NPI on 6/15 | NPI on 6/22 | NPI on 6/29 | NPI on 7/6 |
| 7/1/20 | 24% | 13% | 4% | 0% | 0% |
| 8/1/20 | 65% | 57% | 48% | 38% | 27% |
| 9/1/20 | 82% | 77% | 72% | 65% | 58% |
| 10/1/20 | 84% | 80% | 76% | 71% | 65% |
| 11/1/20 | 82% | 78% | 74% | 69% | 64% |
| 12/1/20 | 78% | 74% | 70% | 66% | 61% |

**Table 3. Percent reductions observed in the hospitalizations by the indicated dates for each NPI initiation date option.**

| | Percent reduction in hospitalizations with NPI initiations | | | | |
|---|---|---|---|---|---|
| by date | NPI on 6/8 | NPI on 6/15 | NPI on 6/22 | NPI on 6/29 | NPI on 7/6 |
| 7/1/20 | 14% | 5% | 1% | 0% | 0% |
| 8/1/20 | 58% | 49% | 39% | 28% | 17% |
| 9/1/20 | 79% | 74% | 68% | 61% | 52% |
| 10/1/20 | 84% | 80% | 75% | 70% | 64% |
| 11/1/20 | 83% | 79% | 74% | 70% | 64% |
| 12/1/20 | 79% | 75% | 71% | 67% | 62% |

September. A similar observation can be made from Table 3; initiation of NPIs on June 22 would also imply 68% reduction in hospitalizations.

Another important insight we gain from the results is the impact that timing of the NPIs makes. In both tables, we see that the reductions by July 1 for the initiation dates of June 29 and July 6 are 0%, since those cases have the same behavior as the baseline 1.67X case. In general, supposing that June 8 would be the earliest that one would trigger an NPI from the time of this modeling exercise, delaying the initiation of NPIs by one week results in about a 5% change in the reductions that we observe in the total infections, hospitalizations and deaths. This assessment model may provide an estimate for the public health burden of each week of delaying policy enactment or individual practice of NPIs.

Another percent reduction metric that one could look at is the maximum number of patients hospitalized under each case. Since the start of the epidemic, the peak hospital resources required to care for COVID-19 patients has been an important concern among public officials indicating the need to flatten the curve. Initiating NPIs on June 8, June 15, June 22, June 29 and July 6 result in respectively 88%, 86%, 84%, 81% and 77% reductions in the maximum number of patients hospitalized (i.e. the peak of each curve). Considering the limitations in the hospital and particularly ICU resources, including expert healthcare personnel, to provide safe and effective care for seriously ill COVID-19 patients as well as patients with critical conditions unrelated to COVID-19, we note that these resource utilization reductions resulting from NPI policy enactment may make a significant difference in population health outcomes.

In addition to the percent reductions in the areas under the total infections and hospitalized curves, we present in Table 4 the percent reductions in the number of deaths by the dates indicated in the first column. Even without considering any negative effects of exceeding hospital care capacity (which is likely to happen without the initiation and widespread adoption of

**Table 4. Percent reductions in deaths by the indicated dates for each NPI initiation date option.**

| | Percent reduction in deaths with NPI initiations | | | | |
|---|---|---|---|---|---|
| by date | NPI on 6/8 | NPI on 6/15 | NPI on 6/22 | NPI on 6/29 | NPI on 7/6 |
| 7/1/20 | 10% | 4% | 0% | 0% | 0% |
| 8/1/20 | 54% | 44% | 34% | 23% | 14% |
| 9/1/20 | 78% | 72% | 66% | 58% | 49% |
| 10/1/20 | 84% | 80% | 75% | 69% | 63% |
| 11/1/20 | 83% | 79% | 75% | 70% | 65% |
| 12/1/20 | 80% | 76% | 72% | 67% | 62% |

NPIs) on patient health outcomes, we see that reductions in deaths resulting from the initiation of NPIs is on the order of 70%. Again, our model demonstrates that one week's delay in the initiation of NPIs corresponds to a 4% to 10% difference in the reductions in deaths.

Further examination of the trends associated with the timing of NPI initiation indicates that the most significant reductions following the successful initiation of an effective NPI is realized within the first two months of policy change. This observation reflects the nature of infection spread in the community for a viral illness with exponential growth potential, as observed with COVID-19. Just as exponential growth experiences a long initiation period in a population prior to sharp rises in case counts, exponential decline in case counts occurs relatively quickly in the population when the transmission routes are attenuated by NPIs, followed by a longer period of more gradual decline that reflects the gradual onset period of the epidemiological curve. It is important to note that the projected reductions in total infections, hospitalizations, and deaths with NPI initiation presume widespread community uptake of any intervention. Poor public adherence to NPI intervention policies are unlikely to result in significant attenuation of viral spread, as policy alone without broad public action is powerless in a pandemic.

## 3 Discussion

In this paper we have proposed a methodology for modelling and projecting the spread of the COVID-19 epidemic in Arizona by considering publicly available data from March 4 (first date with a confirmed COVID-19 case with community spread in Arizona) to June 7. This work is focused on using mathematical modeling techniques to understand the localized features of infection and disease transmission in the early phases of an epidemic, in the absence of extensive available data, as well as exploring the impacts of possible scenarios for implementing control measures through public policy. The description of this work illustrates the ways that epidemiological modeling practices are critically important to the work of public policy-making to promote public health and safety in a pandemic. We note that this model iteration was initially constructed beginning in April at the time of a statewide stay-at-home order, and refined after the stay-at-home order was lifted. This timeframe of the model iteration process allowed for clear observation of the dynamic transmission rates in response to public policy implementation and individual adoption of NPI behaviors, and served as a tool to assist public officials in constructing and implementing policy decisions related to NPIs.

There are several limitations to our analysis. It is important to note that our SEIRD modeling approach did not take into account many factors that play an important role in the dynamics of disease such as heterogeneous contact transmission network, the characteristics of the population (e.g. age, comorbid health conditions, racial and ethnic disparities in access to testing and treatment), the possibility of partial immunity or no immunity from SARS-CoV-2 infection and the availability of testing and contact tracing. At the time of this report, Arizona maintained one of the lowest per capita testing and contact tracing rates of any state in the country. Therefore, it is likely that significant underdetection and thus underreporting of mild and asymptomatic cases may impact calculations of hospitalization and death rates. In order to accommodate this limitation, we used plausible parameters for SARS-CoV-2 based on current evidence. As the evidence related to SARS-CoV-2 and COVID-19 continues to develop, these values are likely to be updated as more comprehensive data become available.

In future work, we look forward to testing this model more broadly against data from other states beyond Arizona in an effort to validate this approach for other public health policy making jurisdictions. In addition, we plan to test this modeling approach more narrowly by applying it to county-specific data in Arizona in order to assess the retrospective accuracy given a

more homogeneous population sample of a single county as opposed to the highly heterogeneous population sample represented by the full state of Arizona. We anticipate that this type of comparative work may inform best practices for early-phase projection modeling in future epidemic conditions. Establishing best practices for early projection modeling can, in turn, provide improved timely inputs for policymakers with more clear expectations and understanding about the scope and limitations of models in highly uncertain conditions like the current COVID-19 pandemic.

Our work illustrates how a system dynamics model can be very useful for making early-phase inferences about how the pandemic impacts may change in response to policy and individual behavior decisions about implementation of different disease mitigation measures.

## Acknowledgments

We would like to acknowledge the contributions of Heidi Gracie and Joshua LaBaer, and the Arizona Department of Health Services Modeling Working Group including Amber Asburry, Steven Robert Bailey, Timothy Flood, Joe Gerald, Katherine Hiller, Ken Komatsu, Mark Manfredo, Vern Pilling, Timothy Richards, Marguerite Sagna, Lisa Villarroel, Patrick Wightman, and Neal Woodbury.

## Author Contributions

**Conceptualization:** Esma S. Gel, Timothy Lant, Heather M. Ross.

**Data curation:** Esma S. Gel, Megan Jehn, Timothy Lant, Anna R. Muldoon, Trisalyn Nelson.

**Formal analysis:** Esma S. Gel, Timothy Lant, Heather M. Ross.

**Investigation:** Timothy Lant, Anna R. Muldoon, Heather M. Ross.

**Methodology:** Megan Jehn, Timothy Lant, Anna R. Muldoon, Trisalyn Nelson, Heather M. Ross.

**Project administration:** Timothy Lant, Anna R. Muldoon, Heather M. Ross.

**Resources:** Timothy Lant.

**Software:** Trisalyn Nelson.

**Supervision:** Timothy Lant, Heather M. Ross.

**Validation:** Esma S. Gel, Timothy Lant.

**Visualization:** Esma S. Gel, Timothy Lant.

**Writing – original draft:** Esma S. Gel, Megan Jehn, Timothy Lant, Anna R. Muldoon, Heather M. Ross.

**Writing – review & editing:** Esma S. Gel, Megan Jehn, Timothy Lant, Anna R. Muldoon, Heather M. Ross.

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
