## [Decision Letter · Decision Letter 0]

23 Jul 2020

PONE-D-20-20416

COVID-19 Healthcare Demand Projections: Arizona

PLOS ONE

Dear Dr. Lant,

Thank you for submitting your manuscript to PLOS ONE. After careful consideration, we feel that it has merit but does not fully meet PLOS ONE’s publication criteria as it currently stands. Therefore, we invite you to submit a revised version of the manuscript that addresses the points raised during the review process.

We look forward to receiving your revised manuscript.

Kind regards,

Akihiro Nishi, M.D., Dr.P.H.

Academic Editor

PLOS ONE

Additional Editor Comments:

Please refer to many of the Reviewers' (mostly Reviewer 1) comments and I am looking forward to seeing your revision.

Journal Requirements:

Reviewers' comments:

Reviewer's Responses to Questions

**Comments to the Author**

1. Is the manuscript technically sound, and do the data support the conclusions?

Reviewer #1: No

Reviewer #2: Yes

2. Has the statistical analysis been performed appropriately and rigorously? 

Reviewer #1: No

Reviewer #2: N/A

3. Have the authors made all data underlying the findings in their manuscript fully available?

Reviewer #1: Yes

Reviewer #2: Yes

4. Is the manuscript presented in an intelligible fashion and written in standard English?

Reviewer #1: No

Reviewer #2: Yes

5. Review Comments to the Author

Reviewer #1: The authors used a compartmental model to describe COVID-19 outbreak in Arizona. Although the overall model structure looks fine, there were some technical issues and inaccuracies. The presentation and organization of the manuscript need further refinement. The simulations were based on very rough assumptions and actual epidemiological relevance is unclear. Substantial revision and/or change of target are recommended for the study to achieve sufficient quality for publication.

Major comments:

- Symptomatic individuals are assumed to remain infectious 6 days after symptom onset, which is not explained in the methods section. This assumption seems inconsistent with the authors' assumption of serial time (serial interval?) of 6 days and lacks empirical evidence (clinically-defined ‘recovery’ does not necessarily correspond to the loss of infectiousness).

- L221: why is it not 5 + 3 days?

- L290: The authors should justify the use of least-square fitting to the number of susceptibles. Fitting to the cumulative data is not recommended as it does not appropriately account for uncertainties in the observed data. Least squares method assumes the data follows a normal distribution with a fixed variance, which may not hold for the epidemic data. The most plausibly, the predicted incidence (considering delay from infection to reporting) should be fitted to the observed case counts (and deaths) using discrete (e.g., Poisson) distributions. Also, most estimates are presented without a measure of uncertainty (confidence intervals).

- Explanation of the model in L170-204 is not well organized. Some epidemiological terms were not appropriately used or defined. Clarity and conciseness are lacking. See also minor comments below. As I also find inaccurate uses of epidemiological terms in other parts of the manuscript, I suggest the author seek for proofreading from colleagues with an epidemiology background.

- Section 1.3: While methodologically being plausible, this initialization method is not novel and is already discussed in a more general model framework of renewal process (for example, see Section 2 of Inaba (2014): https://www.tandfonline.com/doi/abs/10.1080/08898480.2014.891905). Avoid claim of novelty. Moreover, the authors’ approach of simply multiplying and shifting the observed incidence neglects uncertainty around both the delays in infection to reporting and underascertainment. Interpolation shown in Fig. 6 is obviously overfitted. The authors should account for these uncertainties that might have affected their results, especially the confidence intervals.

- Discussion on the ICU cases, mortality rate, and the reported number of deaths is weak in logic. Given the number of confirmed cases and death, the rate of underreporting (“X factor”) and the mortality rate are two completely correlated parameters. The best approach is to fit the model simultaneously to case and death data. If 1.67X scenario does not match the observed number of deaths as L344-345 suggests, this means either the assumption of X = 1.67 or other model settings may be wrong. It did not make sense to me that the authors stick to 1.67X scenario when the model output is inconsistent (mortality > 100%).

- Section 2.1: No citation on the possible “summer effects” on the transmission of SARS-CoV-2? If the authors wish to discuss temperature effects, sufficient background needs to be provided to the readers. Currently, the arbitrary choice of 3 scenarios are not of epidemiological interest since the actual relevance is unknown.

- Section 2.2: I am not sure how useful the “reinitiated NPI” scenarios are when the assumed reduction in transmission cannot contain the epidemic (Reff > 1). Given that the previous state-home-order lasted only for 1.5 months (lifted even before the CDC criteria were met according to Introduction), scenarios assuming the same level of restriction continued for at least 3-6 months (or much longer if continued until the epidemic is over) may not be realistic.

Minor comments

- Figures 1 and 2: It would be helpful if the daily numbers are also shown.

- L102: time from infection to exhibiting infectiousness is called the latent period, not incubation period.

- L168: Equations/through

- L169: The definition of the force of infection is the rate of new infection per susceptible population, which should be -dS/dt/S = βtλ(t) in this model.

- L174-175: It is vague what “these compartments” points to. Also, “patients” usually refers to people visiting hospitals and does not fit in context (presymptomatic individuals are not really patients; “symptomatic patients” suggests they have medically-attended).

- L177-178: the rate at which susceptible individuals get exposed to the infection is βtλ(t), not βt. This sentence is unclear if it means the product of βt, λ(t), and S(t) the authors should rearrange the whole sentence for better readability. Also, the word “exposed to the virus” is vague and should be avoided in this context. “Infectious contact” itself is usually regarded as exposure in epidemiology, whether or not such contact results in infection. Getting into “Exposed” compartment should be termed differently (e.g. become infected).

- L179-180: not clear what is meant to say: “weighted proportion of infectious individuals accounting for the reduced infectiousness of asymptomatic and hospitalized individuals” or something?

- L181-182: this definition of transmission rate is incorrect. As λ(t) represent proportion infectious, βt should be regarded as contact rates x transmission probability, regardless of whether the contactee is infectious or not.

- L191-199: this explanation is redundant and nothing more than rephrasing that βt is a product of contact rate and transmission probability given contact. Figure 4 also seems trivial unless empirical data on the number of contacts in the study population is given. Also, it does not make sense to me that doubling the number of contacts from 10 to 20 does not result in halving the probability of transmission, which instead becomes 0.013 (not 0.01??).

- L262 and thereafter: the word “bin” is misleading. It is “compartment”, isn’t it?

- L268: the term “loading scenario” never explained

- Figure 10: Cumulative curves are not suitable to show the model fit as they tend to look better than it actually is. Use the daily number of deaths instead.

- Section 2.2: The use of the term “NPI” here is inappropriate. NPIs do not only refer to lockdowns or stay-home-orders.

- L420: why is 1.67X “conservative”?

- L490: “proposed a methodology” is an overstatement as this is an application of an existing model to the Arizona data.

- L510: “biologically plausible” Which parameter, and how? I do not see that many biological papers in the reference list.

- L524: I do not fully agree that the model is “locally contextualized” just because the model is fitted to the local data. The model structure is not locally specific and most parameters are cited from the literature.

Reviewer #2: The paper presents a mathematical model to predict the impacts that the COVID-19 epidemic in Arizona will have on the state's healthcare sector. The topic is interesting and worth of investigation. The paper is well organized and the results meaningful. A suggestion to help improve the paper are as follows:

1. I suggest describing the general trend shown in Tables 2, 3, and 4 in greater detail to supplement the current paper's comparison of the varying predictions given different NPI restoration dates.

6. PLOS authors have the option to publish the peer review history of their article (what does this mean?). If published, this will include your full peer review and any attached files.

Reviewer #1: No

Reviewer #2: No

---

## [Author Response · Author response to Decision Letter 0]

7 Sep 2020

Thank you for the detailed review of our manuscript. We appreciate the opportunity to strengthen this work, and have included a revised manuscript along with a detailed response to the suggestions.

---

## [Decision Letter · Decision Letter 1]

14 Sep 2020

PONE-D-20-20416R1

COVID-19 Healthcare Demand Projections: Arizona

PLOS ONE

Dear Dr. Lant,

Thank you for submitting your manuscript to PLOS ONE. After careful consideration, we feel that it has merit but does not fully meet PLOS ONE’s publication criteria as it currently stands. Therefore, we invite you to submit a revised version of the manuscript that addresses the points raised during the review process.

We look forward to receiving your revised manuscript.

Kind regards,

Akihiro Nishi, M.D., Dr.P.H.

Academic Editor

PLOS ONE

Additional Editor Comments (if provided):

While one of the reviewers suggested an accept, the other still does not full agree. Please address the reviewer's comment as much as possible.

Reviewers' comments:

Reviewer's Responses to Questions

**Comments to the Author**

1. If the authors have adequately addressed your comments raised in a previous round of review and you feel that this manuscript is now acceptable for publication, you may indicate that here to bypass the “Comments to the Author” section, enter your conflict of interest statement in the “Confidential to Editor” section, and submit your "Accept" recommendation.

Reviewer #1: (No Response)

Reviewer #2: All comments have been addressed

2. Is the manuscript technically sound, and do the data support the conclusions?

Reviewer #1: Partly

Reviewer #2: (No Response)

3. Has the statistical analysis been performed appropriately and rigorously? 

Reviewer #1: No

Reviewer #2: (No Response)

4. Have the authors made all data underlying the findings in their manuscript fully available?

Reviewer #1: Yes

Reviewer #2: (No Response)

5. Is the manuscript presented in an intelligible fashion and written in standard English?

Reviewer #1: No

Reviewer #2: (No Response)

6. Review Comments to the Author

Reviewer #1: I appreciate the authors’ effort to improve the manuscript for better clarity. The revision did address many of my concerns, while I have a few more comments as bellow on the responses of the authors which I do not fully agree with/believe requires more clarification. Especially, the inconsistent definition of time interval in the model may be a critical point that can change the conclusion of the study.

R1C1-duration of infectiousness: I still don’t know which literature the authors referred to to obtain duration of infectiousness. Ref [6] and [22] are shown in Table 1, but [6] is a modelling study using 6.5 days as generation time (without citing data) and I couldn’t find any mention on the duration of infectiousness in [22]. I would appreciate if the authors could specify which part of the references they cited the duration from.

R1C1-serial interval: It still seems to me that the authors’ explanation that the serial interval is given as 3 days incubation period of a primary case + 3 days incubation period of a secondary case is incorrect. First of all, incubation period (time from infection to “symptom onset”) in this model is not 3 days but 5 days. Even if the authors intend to refer to the latent period (then it will be generation time instead of serial interval), infection at the end of 3 days of the latent period will not correspond to the “mean” serial interval because infection can happen between the beginning and the end of infectious period (hence 3+3 is the shortest possible serial interval). Exact computation of generation time in this model is not straightforward because of the combination of Ip and Is compartment with different infectiousness (but could be done by constructing the survival function of infectiousness: multicompartment-equivalent of Eq 6 in Inaba 2014), but it is at least 3 (latent) + 6 (mean duration of Is) + α (duration of Ip modified by lower infectiousness). I believe that the mean serial interval being longer than 3 days could also be checked by simulating this model with explicit tracking of generations. This inconsistency in the model setting should be addressed.

R1C1 “Based on the literature for COVID-19, we assumed that an individual’s case starts at the time that the individual becomes presymptomatic (and infectious) and that individuals are fully infectious during this presymptomatic period, which is modeled to be about 2 days in duration.”:

I’m not sure what “individual’s case” refers to, but serial interval is specifically defined as time between symptom onsets of linked cases, in which sense the transition from Ip to Is should be regarded as an onset. Also, “individuals are fully infectious during the presymptomatic period” sounds inconsistent with the model assumption that those in Ia compartment has infectiousness 0.55 times that in Is, according to the definition of λ.

R1C3-fitting target: If the least-squares fitting was used for the daily counts of cases, I believe the description between L292-298 should be rewritten because it is misleading. It is not indicated that the daily count was used; moreover, showing the number of susceptibles in the context of fitting implies that it might have been the target of fitting.

R1C3-discrete distribution: I understand that the ODE solution gives noninteger values, but this does not prohibit the use of discrete distributions such as Poisson distribution (mean of Poissson distribution can be noninteger). The central limit theorem assures approximate normality if the process is simulated many times as the authors suggest in the response, but because the available data is only one realization of the process and not the average of many simulations, this does not necesarily apply. Instead, the central limit theorem may ensure that the Poisson model is approximately normal (if the mean is large enough), but then the least-squares relative error method should be used instead of absolute error (because Poisson(µ)≃Norm(µ, µ)).

R1C5-reference to Inaba 2014: Apologies for my comment not specific enough, I wanted to especially highlight Eq 5 in Section 2, which shows exactly what is done as initialization in the current paper. Note that the model here is a generalization of compartmental models and includes the SEIRD model as a special case. The second line of Eq 5 represents the “backcasting” of the past states of compartments (y(t,τ,ξ) in a general form), while the first line predicts the onward transmission.

R1C7-Background of summer effects: I understand that the evidence at the earlier time was mixed and the authors may have had to draw scenario projections based on rough assumptions. Still, I believe it would be useful to provide the readers with context by clarifying what data on summer effects was available at the time of study (I believe at least some preliminary data or reasoning existed that hypothesized the summer effects), and how our understanding has changed since then.

Reviewer #2: (No Response)

7. PLOS authors have the option to publish the peer review history of their article (what does this mean?). If published, this will include your full peer review and any attached files.

Reviewer #1: No

Reviewer #2: No

---

## [Author Response · Author response to Decision Letter 1]

11 Oct 2020

Please see the attached response to the reviewers document for responses to all of the comments. Thank you.

---

## [Decision Letter · Decision Letter 2]

20 Oct 2020

PONE-D-20-20416R2

COVID-19 Healthcare Demand Projections: Arizona

PLOS ONE

Dear Dr. Lant

Thank you for submitting your manuscript to PLOS ONE. After careful consideration, we feel that it has merit but does not fully meet PLOS ONE’s publication criteria as it currently stands. Therefore, we invite you to submit a revised version of the manuscript that addresses the points raised during the review process.

We look forward to receiving your revised manuscript.

Kind regards,

Akihiro Nishi, M.D., Dr.P.H.

Academic Editor

PLOS ONE

Additional Editor Comments (if provided):

The reviewer #1 wants to see some improvements. Please address them for the acceptance.

Reviewers' comments:

Reviewer's Responses to Questions

**Comments to the Author**

1. If the authors have adequately addressed your comments raised in a previous round of review and you feel that this manuscript is now acceptable for publication, you may indicate that here to bypass the “Comments to the Author” section, enter your conflict of interest statement in the “Confidential to Editor” section, and submit your "Accept" recommendation.

Reviewer #1: (No Response)

Reviewer #2: All comments have been addressed

2. Is the manuscript technically sound, and do the data support the conclusions?

Reviewer #1: Yes

Reviewer #2: (No Response)

3. Has the statistical analysis been performed appropriately and rigorously? 

Reviewer #1: Yes

Reviewer #2: (No Response)

4. Have the authors made all data underlying the findings in their manuscript fully available?

Reviewer #1: (No Response)

Reviewer #2: (No Response)

5. Is the manuscript presented in an intelligible fashion and written in standard English?

Reviewer #1: Yes

Reviewer #2: (No Response)

6. Review Comments to the Author

Reviewer #1: Thanks to the authors’ clarification I believe the manuscript has been greatly improved and almost ready to be published. I only have two more comments.

- Regarding “R1C3-discrete distribution”: What I was concerned with was what data generating process the authors assumed for the observed data. From the responses, it seems that the authors did not assume any stochastic uncertainty in the relationship between the “presumed exposures” and the daily case counts and simply estimated the ODE which best describes the presumed exposures in terms of squared errors. I do not think this is considered to be the standard handling of statistical uncertainty: given the presumed “exposures”, the observed data as case counts should follow some sort of counting process which involves stochastic fluctuation. I proposed the use of Poisson process to (at least partially) account for this stochastic nature of the observation. In the current implementation, the authors are essentially making an assumption that they can obtain the presumed exposure deterministically from the observed case counts. As a result, the uncertainty in the result (CI in estimates) are underevaluated. However, as the overall implication might not be largely affected, I would understand if the authors just acknowledge the limitations and possible effects of the use of deterministic process in the discussion.

- Claim of novelty: thank you for clarifying what the authors meant by “novelty”. If “X-factoring” alone constitutes the novelty as you mention, Inaba (2014) is indeed irrelevant. My understanding is Inaba (2014) mathematically explains how previous incidence can be obtained from the initial condition (i.e. the current observation of cases) and thus is a generalization of the “initialization approach”. Note that this paper is not intended to claim the novelty of this approach, but is just using it as a well-known existing concept. Regarding “X-factoring”: this approach is already familiar to infectious disease epidemiologists in a different form for an adjustment of underreporting. Often this is handled by assuming the reporting probability p and incorporate binomial reporting process as part of the model, instead of inflating the observed case counts, to appropriately account for uncertainty in reporting. I do not believe that the authors should claim the novelty of X-factoring just because it’s not often used in previous studies because the method is subject to a strong assumption (negligence of stochastic fluctuation). I think the claim of novelty here is not really necessary for this paper to stand and the authors can simply drop the claim without losing its value.

If these two points have been addressed, I am happy for this paper to be published.

Reviewer #2: (No Response)

7. PLOS authors have the option to publish the peer review history of their article (what does this mean?). If published, this will include your full peer review and any attached files.

Reviewer #1: No

Reviewer #2: No

---

## [Author Response · Author response to Decision Letter 2]

2 Nov 2020

Please see attached document for response to reviewer's comments.

---

## [Decision Letter · Decision Letter 3]

6 Nov 2020

COVID-19 Healthcare Demand Projections: Arizona

PONE-D-20-20416R3

Dear Dr. Lant,

We’re pleased to inform you that your manuscript has been judged scientifically suitable for publication and will be formally accepted for publication once it meets all outstanding technical requirements.

Kind regards,

Akihiro Nishi, M.D., Dr.P.H.

Academic Editor

PLOS ONE

Additional Editor Comments (optional):

NA

Reviewers' comments:

Reviewer's Responses to Questions

**Comments to the Author**

1. If the authors have adequately addressed your comments raised in a previous round of review and you feel that this manuscript is now acceptable for publication, you may indicate that here to bypass the “Comments to the Author” section, enter your conflict of interest statement in the “Confidential to Editor” section, and submit your "Accept" recommendation.

Reviewer #1: All comments have been addressed

2. Is the manuscript technically sound, and do the data support the conclusions?

Reviewer #1: (No Response)

3. Has the statistical analysis been performed appropriately and rigorously? 

Reviewer #1: (No Response)

4. Have the authors made all data underlying the findings in their manuscript fully available?

Reviewer #1: (No Response)

5. Is the manuscript presented in an intelligible fashion and written in standard English?

Reviewer #1: (No Response)

6. Review Comments to the Author

Reviewer #1: (No Response)

7. PLOS authors have the option to publish the peer review history of their article (what does this mean?). If published, this will include your full peer review and any attached files.

Reviewer #1: No

---

## [Editor Report · Acceptance letter]

17 Nov 2020

PONE-D-20-20416R3 

COVID-19 healthcare demand projections: Arizona  

Dear Dr. Lant:

I'm pleased to inform you that your manuscript has been deemed suitable for publication in PLOS ONE. Congratulations! Your manuscript is now with our production department. 

Kind regards, 

on behalf of

Dr. Akihiro Nishi 

Academic Editor

PLOS ONE